



# The Influence of Large Woody Debris on Post-Wildfire Debris Flow Sediment Storage

Francis K. Rengers [1], Luke A. McGuire [2], Katherine R. Barnhart [1], Ann M. Youberg [3], Daniel Cadol [4], Alexander N. Gorr [2], Olivia Hoch [1], Rebecca Beers [3], and Jason W. Kean [1]

[1]U.S. Geological Survey, Geologic Hazards Science Center, Golden, CO
[2]University of Arizona, Tucson, AZ
[3]Arizona Geological Survey, Tucson, AZ
[4]New Mexico Tech, Soccoro, NM

**Correspondence:** Francis K. Rengers (frengers@usgs.gov)

**Abstract.** Debris flows transport large quantities of water and granular material, such as sediment and wood, and this mixture can have devastating effects on life and infrastructure. The proportion of large woody debris (LWD) incorporated into debris flows can be enhanced in forested areas recently burned by wildfire because wood recruitment into channels accelerates in burned forests. In this study, using four small watersheds in the Gila National Forest, New Mexico, that burned in the 2020

5 Tadpole Fire, we explored new approaches to estimate debris flow velocity based on LWD characteristics and the role of LWD on debris flow volume retention. To understand debris flow volume model predictions, we examined two models for debris flow volume estimation: (1) the current volume prediction model used in U.S. Geological Survey debris flow hazard assessments, and (2) a regional model developed to predict the sediment yield associated with debris-laden flows. We found that the regional model better matched the magnitude of the observed sediment at the terminal fan, indicating the utility of regionally calibrated

10 parameters for debris flow volume prediction. However, large wood created sediment storage upstream from the terminal fan, and this volume was of the same magnitude as the total volume at the terminal fans. Using field and lidar data we found that sediment retention by LWD is largely controlled by channel reach slope and a ratio of LWD length to channel width between 0.25 and 1. Finally, we demonstrated a method for estimating debris flow velocity based on estimates of the critical velocity required to break wood, which can be used in future field studies to estimate minimum debris flow velocity values.

## 1 Introduction

It has long been recognized that large wood influences a variety of hydraulic, ecologic, and sediment transport processes in fluvial systems (e.g., Swanson and Lienkaemper, 1978; Abbe and Montgomery, 1996; Montgomery et al., 2003a; Wohl, 2013). For example, large woody debris (LWD) ($> 10$ cm diameter and $> 1$ m length; Comiti et al. (2016)) creates habitat complexity and functionality for ecological services in flowing streams (e.g., Montgomery et al., 2003b; Vaz et al., 2013). In addition,

20 channel reaches with LWD tend to retain sediment for longer periods of time (Faustini and Jones, 2003; Grabowski and Wohl, 2021) because LWD increases the sediment storage capacity (e.g., Keller and Swanson, 1979; Megahan, 1982; Montgomery et al., 1996; May and Gresswell, 2003) and dissipates energy, encouraging deposition (Richmond and Fauseh, 1995). Wood





can be introduced into channels by mass movement such as landslides, streambank failure, and debris flows (Swanson and Lienkaemper, 1978; Montgomery et al., 2003b; May and Gresswell, 2003; Wohl et al., 2009; Chen et al., 2013; Lucía et al., 2015; Surian et al., 2016). Moreover, forests burned by wildfire often accelerate the introduction of wood into channels through processes such as root weakening, wind throw, and disease (Benda et al., 2003; Zelt and Wohl, 2004; Chen et al., 2005; Jones and Daniels, 2008; Bendix and Cowell, 2010).

The interaction between debris flows and LWD is complex because debris flows both scour (e.g., May and Gresswell, 2003; Vascik et al., 2021) and deposit LWD (e.g., Montgomery et al., 2003b; May and Gresswell, 2004) in channels. LWD scoured by debris flows can remain entrained for the full runout zone of a debris flow (Booth et al., 2020). In cases where large amounts of LWD are moving within a debris flow, the runout length tends to be shorter than in debris flows with less LWD (Booth et al., 2020; May and Gresswell, 2004). Shorter runout distances have been attributed to the jamming effects of wood (Booth et al., 2020), and modeling has shown that large wood entrained in debris flows can reduce the flow velocity, which could further reduce the runout distance (Lancaster et al., 2003). In some debris flows, LWD can be deposited at various points along the runout path. Changes in the local geomorphology (e.g., slope change at tributary junctions or channel width) can encourage LWD deposition, and LWD can also be stopped by in-channel immobile objects (e.g., standing trees, large boulders), creating barriers that retain upstream sediment (Montgomery et al., 2003b; May and Gresswell, 2004; Lancaster and Grant, 2006).

From a hazards perspective, the incorporation of LWD in debris flows poses a threat to human life and infrastructure (e.g., Comiti et al., 2016). Damage to roads, bridges, and reservoirs from large wood transport has been documented during flood events (Shrestha et al., 2012; Lucía et al., 2015; Surian et al., 2016; Steeb et al., 2017; Piton et al., 2020), and the majority of wood supplied to the floods originated from landslides and debris flows in low-order drainages or on hillslopes (Chen et al., 2013; Lucía et al., 2015; Surian et al., 2016; Comiti et al., 2016; Rathburn et al., 2017). LWD has also been shown to support sediment retention in landslide dams (Struble et al., 2021). These large wood debris jams can break catastrophically (Swanson and Lienkaemper, 1978; Coho and Burges, 1994; Abbe and Montgomery, 2003), sending a destructive wave of debris and water downstream. The depth of flow moving downstream is amplified above water-only flow by the sediment and debris, making these flows more destructive (Kean et al., 2016).

In this study, we examined the role of LWD on sediment storage of debris flow deposits. Understanding and predicting the volume of debris flow deposition in a watershed is important for hazard assessment. Studies have used debris flow or debris laden flood observations to develop predictive models of sediment transported after wildfires (Gartner et al., 2014; Pelletier and Orem, 2014; Nyman et al., 2015; Rengers et al., 2021). Although these models implicitly include any sediment retention by LWD, the current predictive approaches do not explicitly account for the sediment storage potential created when wood self-organizes in a channel to block the upstream debris flow sediment. Therefore, we explored the ability of LWD to store debris flow sediment after several runoff-generated debris flows following the 2020 Tadpole Fire in the Gila National Forest in New Mexico, USA. In this study, we specifically investigated how LWD characteristics (e.g., diameter, length, class) influenced the deposit volume that was retained. In addition, we explored relations between the deposit volume retained by LWD and the local geomorphic characteristics (e.g., channel slope, channel width, drainage area). Through this work we are able to better understand and anticipate how LWD may control debris flow sedimentation, and we established a new approach for estimating





the critical velocity required to break wood in order to back-calculate debris flow velocity in forested settings. Understanding the flow velocity helps to constrain model estimates of debris flow runout (Barnhart et al., 2021) and building damage (Kean et al., 2019).

## 2 Study Site

The Tadpole wildfire started on 6 June 2020 in the Gila National Forest and burned 45 square kilometers before containment in July 2020. The pre-fire vegetation was dominated by Ponderosa Pine (*Pinus ponderosa*). The local geology is composed of Tertiary-aged volcanic rocks, pyroclastic rocks, and ash flow tuffs of the Datil Group (Scholle, 2003), and the dominant soils on the site are Mollisols, Enceptisols, and Alfisols (U.S. Forest Service, 2020). This study focused on debris flows that initiated near the crest of Tadpole Ridge in four watersheds burned primarily at moderate-to-high severity (Figure 1 and Table 1). The study area falls within a semi-arid climate with annual rainfall totals from 40 to 100 cm, and rainfall occurs primarily during the summertime as part of the North American Monsoon (Bonnin et al., 2004).

At this study site, abundant woody debris was available on the forest floor after the wildfire, as well as trees that remained upright after burning. The large diameter ponderosa pine woody debris is unlikely to be fully consumed during short duration fires, as the consumption of wood is related to wood diameter. For example, round-diameter deadwood 1-hour fuels are $<=$ 0.64 cm, 10-hour fuels are 0.64-2.5 cm, and 100-hour fuels are 2.5-7.6 cm (National Wildfire Coordinating Group, 2022). Because wildfire duration is typically less than 10-100 hours, we expect large diameter wood (e.g., $>$ 10 cm) to remain in the forest and to be available to interact with channel sediment after wildfires. Moreover, Ponderosa Pine wood diameters $>$ 10 cm differentiate this site from locations such as the San Gabriel Mountains in California, where the maximum diameter of chaparral plant stems fall between the 1-hour and 10-hour fuel diameters (Conard and Regelbrugge, 1994) and are often fully consumed during a fire.

## 3 Methods

The following subsections outline the methods used in this study, including in situ field instrumentation, airborne lidar, and field mapping. We also describe the analytical methods used to compare volume measurements with existing empirical volume models. Finally, we outline a new method for using wood observations to estimate flow velocity.

### 3.1 Instrumentation, Mapping, and Measurements

We installed equipment to monitor runoff and debris flow responses from four watersheds on 6-7 July 2020, while portions of the Tadpole wildfire were still burning (Tad-1, Tad-2, Tad-3, and Tad-4, Figure 1). Monitoring equipment for this study was clustered at two locations: one location included a stand-alone rain gauge (RG), and a second location included a rain gauge with paired geophones to record the timing and velocity of debris flows (RG & Geophones) (Figure 1). The geophones (single-component, Geospace GS11) were emplaced into the ground using a spike that contacted the soil. They were programmed to





only turn on during rainfall, and they recorded at a rate of 50 $Hz$. At the geophone location, the geophones were aligned on the channel bank parallel to the channel (Figure 2). The geophones were located a horizontal distance of 13.9 m from the channel

edge, a vertical distance of 3.1 m above the channel thalweg, and they were spaced 14.6 m apart (Figure 2).

In the watershed monitored using geophones, we estimated the debris flow velocity by using a cross-correlation analysis to estimate the time difference between the absolute value of the two geophone measurements [$mV$] (data available in Rengers et al. (2022a)). Additionally, we filtered the signal using a 5-second median filter and divided each geophone signal [$mV$] by the maximum value during the storm. Using this instrumentation, in addition to field visits, we were able to identify post-wildfire

rainstorms that resulted in runoff-generated debris flows.

We mapped debris flow deposits in four watersheds using ArcGIS Collector (Figure 1). The volume of each debris flow deposit was estimated as a sediment wedge using a measuring tape, similar to the approach described by Lancaster et al. (2003). Photographs were obtained at each deposit location and attached to the Collector points (data available in Rengers et al. (2022b)).

In the same four watersheds, we also mapped the LWD in channels. The LWD was classified using terminology borrowed from the fluvial literature to describe mapped wood as: buried, loose, ramp, bridge, or jam (Figure 3) (Kramer and Wohl, 2017). Buried LWD is defined here as wood that is contained within and underneath sediment. Buried LWD can also be pinned by a tree, boulders, or other wood in a wood jam. When wood is pinned, debris flow sediment pushes the wood against a large object with enough resistance to keep the wood in place. In these cases the buried wood can help to retain sediment within a

channel such that, without the buried/pinned wood, it is unlikely that sediment would have deposited at that specific location in the channel (Figure 3). By contrast, loose LWD is stratigraphically on top of a sediment deposit or the channel. Loose pieces can float during water flows or become pinned by downstream trees, boulders, or jams, but they do not actively retain any sediment. Bridge LWD are wood pieces that are longer than the channel width and therefore span the channel banks, often not interacting with the channel flow or sediment. Ramps are loose LWD that have fallen into the channel, and a portion of the

LWD remains on the channel bank. Ramps can be pinned by downstream obstacles or partially buried to retain sediment in the flow, but they protrude out of the active channel. Finally, jams are composed of many pieces of LWD interlocked via friction that block a portion of the channel.

We used pre-event high-resolution topographic data to explore the connection between the geomorphology of the debris flow producing watersheds and the deposits forced by LWD. Airborne lidar data were flown prior to the fire on 27-28 March 2019

with a point density of 4.9 points/m$^2$. We obtained the lidar point clouds from the National Map (U.S. Geological Survey, 2019), and stitched them together using LAStools (Rapidlasso, 2022) in order to create a hydrologically connected digital elevation model (DEM). We used this pre-fire lidar data from the study site to compare the length of the LWD to the pre-debris flow channel width, and we examined the relationship between pre-event slope and deposit volume.

In order to extract the channel width, we first defined the stream channels using the hydrologic toolset in ArcGIS Pro 2.8.3,

calculating the upstream contributing drainage area using a D8 flow-direction algorithm. This flow accumulation grid was subsequently used to identify the stream network with a threshold contributing area of 0.01 km$^2$. We then created stream cross-sections perpendicular to the stream channels at a spacing of 5 m, and we associated each debris flow deposit to the





nearest cross-section. For each cross-section near a measured deposit, we extracted the x, y, z values from the DEM underlying

each cross-section. Using the cross-sectional profile (Figure 4a), we defined the active channel width as the flow width 1

m above the lowest elevation in the channel, which corresponded to the peak flow depth observed in most channels during

field observations. Topographic measurements of the active channel width derived from lidar were compared against field

measurements of the active channel width.

  Prior work has recognized that LWD deposition in a channel is related to the length of the LWD versus the channel width

(Vaz et al., 2013). Consequently, we examined the volume of debris flow sediment stored behind LWD with respect to the ratio

of wood length ($L$) to channel width ($W$) using:

$$\zeta_{LW} = \frac{L}{W} \tag{1}$$

In addition, buried, jam, and ramp LWD classes were influential in storing sediment when they are pinned against an object

such as a tree or large rock that did not move in the flow. Therefore, we accounted for whether LWD was pinned at a location

of sediment deposition. In this analysis we eliminated all LWD measurement locations where the deposit volume stored was 0

m$^3$. This eliminated all of the bridges and loose LWD from the analysis because neither led to the storage of sediment.

### 3.1.1 Volume Models

The debris flow deposit volumes forced by LWD retention in our study area were further compared to modeled predictions of

debris flow volume. The U.S. Geological Survey (USGS) debris flow hazard assessment uses a model developed by Gartner

et al. (2014) in the Transverse Range of southern California to estimate debris flow volumes. The volume model has the

following form:

$$\ln(V) = 4.22 + 0.39\sqrt{I15} + 0.36 ln(Bmh) + 0.13\sqrt{R} \tag{2}$$

where $V$ is volume (m$^3$), $I15$ is the 15-minute rainfall intensity (mm/hr), $Bmh$ is watershed area burned at moderate and

high severity (km$^2$), and $R$ is the watershed relief (m). The model was developed in an area dominated by chaparral shrub

forests and scrub oak vegetation at elevations below 1520 m, and conifer forests above 1520 m with Douglas-fir (*Pseudotsuga*

*macrocarpa*), coast Douglas-fir (*Pseudotsuga menziesii var. menziesii*), ponderosa pine (*Pinus ponderosa*), white fir (*Abies*

*concolor*), and lodgepole pine (*Pinus contorta*) (U.S. Forest Service, 2022). Because of the large swaths of chaparral, the

availability of large wood able to retain debris flow sediment is reduced compared to forests with larger trees, such as the

Tadpole study site. This model is applied to channel segments modeled as part of a USGS debris flow hazard assessment

(U.S. Geological Survey, 2022), and values for $\ln(Bmh)$ and $\sqrt{R}$ are calculated for each segment. For the rainfall intensity

parameter in Equation 2, we used the gauge labeled RG & Geophones (Figure 1) for drainages Tad-1 and Tad-2, and the gauge

labeled RG (Figure 1) for drainages Tad-3 and Tad-4. For this calculation, we used the maximum observed $I15$ at each of the

rain gauges for the highest intensity rainstorm on 8 September (Table 2). This was not the only debris flow producing storm,

but it was the storm with the highest intensity and therefore the calculated volumes would show maximum potential volume.





In addition to the Gartner et al. (2014) model, we used a model developed in New Mexico to predict sediment yield associated
with debris-laden flows to compare with our observations (Pelletier and Orem, 2014):

$$Y_p = aS^b B^c \tag{3}$$

where $Y_p$ is sediment yield in mm, $S$ is average basin slope (m/m), $B$ is average soil burn severity, $a = 1.53$, $b = 1.6$, and $c = 1.7$.
The coefficients used in Equation 3 could be calibrated to any regional setting; however, because this study site is near the area
that the model was developed, we used the original coefficients. The sediment yield $Y_p$ was converted from units of millimeters
to meters and multiplied by the upstream basin area in order to obtain a volume. The debris flows documented by this model
originated in areas with large trees and deposited in a fan dominated by grass and shrubs (Pelletier and Orem, 2014). Pelletier
and Orem (2014) describe the vegetation of their study area as ponderosa pine and Gambel oak (*Quercus gambelii*) below
2740 m, Douglas fir (*Pseudotsuga menziesii*), white fir (*Abies concolor*), blue spruce (*Picea pungens*), and aspen (*Populus
tremuloides*) stands between 2740 and 3040 m, and Engelmann spruce (*Picea engelmannii*) and corkbark fir (*Abies lasiocarpa
var. arizonica*) > 3040 m.

The volume models are expected to predict the total volume passing a location, whereas the total volume retained behind
LWD only reflects the amount of material that was stopped by LWD. Consequently, we compared the maximum modeled
volume with the volume of terminal fans in Tad-1-3. Tad-4 did not have a terminal fan at the basin outlet, so the nearest deposit
behind LWD was used. For additional context, we compared the total volume stored behind LWD upstream from the terminal
fan with the fan deposits and modeled fan volumes.

### 3.2  Velocity Estimates from Wood Measurements

Understanding breaking forces/velocities may help to identify the threshold where LWD substantially influences exported
sediment volumes. Therefore, we related wood size to breaking velocity, which is the velocity of flow required to break wood,
assuming greenstick fracture behaviour (Ennos and Van Casteren, 2010). Wood transported in a debris flow experiences large
forces that may splinter or break the wood into smaller fragments. The size of wood remaining after a debris flow event may
provide constraints on the debris flow velocity, in that velocities in excess of an estimated breaking velocity were likely not
experienced. The estimated breaking velocity should thus be considered as the flow velocity threshold necessary to break
LWD. To estimate the breaking velocity, we consider the wood as a cylindrical beam with length $L$ and diameter $D$ that is
pinned either by two downstream trees—one at each end—or by one downstream tree located at the midpoint, $L/2$. The beam
is subjected to a uniform force per unit length $f$ directed in the downstream direction. This uniform force bends the LWD,
imparting a maximum bending moment $M$, in both idealized geometries, occurring at the mid-point of the LWD piece.

$$M = \frac{1}{8} f L^2 \tag{4}$$

A complete description of $f$ would necessitate describing the depth-variable flow field of the debris flow front, including
the force of impact imparted by entrained boulders. As a rough approximation, we considered only the force imparted by fluid





drag. We assumed that the LWD piece was fully submerged, and that flow was both above and below the tree in order to calculate the total drag force $F$ as:

$$F = \frac{1}{2}\rho u^2 C_d A \tag{5}$$

where $u$ is the downstream velocity, $\rho$ is the fluid density, $C_d$ is the drag coefficient, and $A$ is the cross-sectional area facing the flow ($A = D * L$). We used $\rho = 1680$ kg/m$^3$ reflecting a solid volume fraction of 0.6 and a sediment density of 2700 kg/m$^3$.

We used a large value for $C_d = 1.17$ corresponding to a submerged cylinder. The force per unit length is given as:

$$f = \frac{F}{L} = \frac{1}{2}\rho u^2 C_d D \tag{6}$$

Following (Ennos and Van Casteren, 2010) (their Equations 1.6 and 2.8) we calculated both the maximum longitudinal and transverse stresses, $\sigma_L$ and $\sigma_T$, respectively, within the LWD piece for a range of $D$ and $L$.

$$\sigma_L = \frac{32M}{\pi D^3} \tag{7}$$


$$\sigma_T = \frac{1024M^2}{3\pi^2 D^6} \tag{8}$$

As discussed by Ennos and Van Casteren (2010), LWD pieces and other natural beams are stronger in the longitudinal direction (parallel to $L$) and typically break in the transverse direction. Accordingly, we used Equations 3.3 and 3.4 from Ennos and Van Casteren (2010) to calculate the longitudinal and transverse yield strengths, $\sigma_{Ly}$ and $\sigma_{Ty}$, respectively, assuming a tree

density of 500 kg/m$^3$ (Engineering Toolbox, 2022). We then calculated the value of $u$ at yield, the lowest velocity for which the maximum stress in either the longitudinal or transverse direction is equivalent to the relevant yield strength.

We assumed failure always occurred in the transverse direction, associated with the following equation for breaking velocity

$$u = \frac{D}{L}\left(\frac{\pi}{2\rho C_d}\right)^{0.5}(3\sigma_{Ty})^{0.25} \tag{9}$$

implying that for a constant ratio of $D/L$ the breaking velocity is constant. We then used field measurements of $D$ and $L$ to estimate $u$ at locations where LWD was pinned against trees.

## 4    Results

There were 11 substantial rain events during the summer monsoon period following the wildfire (Table 2). These resulted in multiple runoff-generated debris flows in each basin, and in all of the basins it was possible to identify the date of the largest

debris flow (Table 1). Field observations indicate that LWD storage of debris flow material occurred during all of these storms,





and thus the measurements represent an aggregate across debris flow events. Note that the terminal deposit in Tad-4 was eroded prior to measurement. The largest recorded storm occurred on 8 September 2020 (Table 2), and the geophones during that storm provided the clearest estimate of debris flow velocity (Figure 5). Using the data from this storm, we found the maximum lag between the upstream and downstream geophone peaks was 3.6 seconds, indicating a debris flow velocity of 4.1 $m/s$ (Figure
215  5).

We mapped 218 locations of LWD within the four study watersheds, which were associated with 124 unique debris flow deposits (some deposits had more than 1 piece of LWD) (Figure 1). The total volume stored by each LWD class shows that the buried and jam LWD classes were associated with the largest cumulative deposit volume stored (Figure 6). Buried and jam LWD at the field site were often pinned against stable objects such as standing trees or boulders, and the buried wood pieces
created a barrier that retained an upstream sediment deposit (Figure 6). Loose wood was also found in debris flow deposits, possibly deposited during the waning watery tail of debris flows, but loose wood did not provide any structural stability that would retain the deposit (Figures 3 and 6). Finally, ramps were associated with the smallest deposit volumes.

We compared the maximum measured LWD diameter of buried and jam LWD classes to the deposit volume (Figure 6d-f), limiting our analysis to these two classes because they were associated with the largest deposits (Figure 3). LWD with
diameters larger than 20 cm were associated with larger sediment deposits. Additionally, as the LWD diameter for these two classes increases, the number of observations decreases, but the total stored volume per number of deposits increases (Figure 6d). For example, 50% of the observed sediment volume was retained behind LWD with a maximum diameter between 20 and 30 cm, but only 30% of the maximum measured diameters are between 20 and 30 cm (Figure 6d). Similarly, 16% of the observed volume was stored behind a maximum LWD diameter of 50-60 cm, but those maximum diameter sizes only
represented 7% of the total measured diameters (Figure 6d-f).

The ratio of LWD length to channel width ($\zeta_{LW}$) also influenced the volume of trapped debris flow sediment. The maximum debris flow deposit volumes were concentrated within a narrow range of $0.25 < \zeta_{LW} < 1$ (Figure 7). In the majority of the measurements, LWD was pinned by a larger immobile downstream object, causing sediment to backup behind the LWD. Among the different LWD classes, ramps did not stop a large amount of sediment (Figure 3), but they span a large range from
$\zeta_{LW}$ less than 1 to $\zeta_{LW}$ greater than 1. Because many ramps were buried, the true LWD length was likely underestimated in those cases, thus contributing to estimates of $\zeta_{LW} < 1$. The peak in sediment retention in the range of $0.25 < \zeta_{LW} < 1$ reflects situations where the wood is small enough to fit in the channel, unlike a bridge, but large enough to take up a large proportion of the channel width where the wood could be wedged between standing trees or boulders within the channel to stop upstream sediment. Buried LWD and jams were the primary classes of LWD associated with ratios between 0.25 and 1, containing the
majority of larger deposit volumes (e.g., $> 10\,\mathrm{m}^3$).

The analysis of sediment volume with respect to channel slope showed no strong relationship between measured volume and slope if all of the measured deposits were considered. However, lower channel slope is correlated with sediment volume above a threshold size of $10\,\mathrm{m}^3$ (Figure 8). The largest debris flow deposits were observed where the local pre-fire channel slope was $> 5°$ and $< 25°$. No post-fire slope data are available in cases where the channel slope may have changed, but qualitative field



observations indicate that source areas scoured to bedrock and steepened and depositional areas aggraded creating shallower slopes.

    The total volume stored behind LWD was larger than the volume of sediment stored in terminal fans for most of the drainages (Tad-2-4) and comparable in size in Tad-1 (Figure 9). The Gartner et al. (2014) volume model overpredicted the volume of the observed terminal fans by 1-4 orders of magnitude, whereas the Pelletier and Orem (2014) model provided estimates that were
nearly same magnitude as the terminal fan observations (excluding Tad-4 where the terminal fan was removed by erosion).

    Our wood-break analysis showed that the velocity required to break wood (in the considered idealized geometry) varied across wood lengths and diameters. We found clear spatial patterns of velocity by applying the peak velocity from the largest rainstorm on the study site with LWD that was either buried or in a jam. In Tad-1, where the field velocity measurement was made, we found that the measured LWD in the channel is all larger than the wood geometry that would be broken by a velocity
of 4 m/s, with the exception of the LWD at the bottom of the watershed where the channel widens and debris flow sediment is deposited in a fan (Figure 10). This result agrees well with the breaking velocity approach. In the other drainages (Tad-2-4) wood with $D/L$ measurements below the breaking velocity was primarily located in wide channel reaches where velocity would be expected to slow, otherwise the wood geometry is consistently larger than the modeled breaking velocity (Figure 10).

## 5   Discussion

This study examines how post-fire debris flows moving through small headwater channels in forested environments retain debris flow sediment, where debris flow sediment is stored, and the geomorphic/wood characteristics that influence local deposition. Field data combined with modeling are used to understand how large woody debris influences debris flow volume storage and velocity. Better constraints on sediment volume and velocity will ultimately lead to more accurate debris flow runout modeling and damage assessments (Kean et al., 2019; Barnhart et al., 2021).

Field measurement data indicate that wood characteristics played an important role in the depositional volume and location. For example, the maximum diameter of LWD in a channel reach was related to the total deposit volume stored. The majority of wood diameters measured were greater than 10 cm, possibly because wood of smaller diameters were fully consumed by the fire, considering that 10-100 hour fuels are (2.5-7.6 cm). Consequently, in forest environments with smaller diameter wood (e.g., chapparal) the effect of wood on sediment storage may be limited compared to forests with larger trees. Moreover, the
class of the LWD strongly influenced deposit volume storage, with LWD that was buried or in a jam containing the most sediment (Figure 3). Ramps retained little sediment, likely because they are unstable features.

    The combined influence of channel morphology and LWD also influenced debris flow sediment storage. The ratio of wood length to channel width ($\zeta_{LW}$) shows that sediment is preferentially stored where LWD spans at least one quarter of the channel width. When LWD is longer than the channel width, it does not effectively stop sediment because it cannot be oriented fully
perpendicular to the flow field. In cases where the LWD is less than one quarter of the channel width, the flow can move around or reorient the LWD, making the wood less effective for storage (see Figure 3b for an example of sediment storage in half the channel width, and flow movement through the other half of the channel.). Identifying critical reaches of potential





sediment storage might be done a priori using channel width measurements from high-resolution topography, if a characteristic LWD length could be estimated in a region. Prior work has shown that channel width increases as a function of discharge

(e.g., $w = cQ^b$) (Leopold et al., 1964). However, in the steep headwater catchments in our study area, width does not appear to change predictably as a function of drainage area (Figure 4c). Instead, narrow local channel reaches allowed LWD to deposit and then store debris flow sediment moving from upstream. Our comparison of field-measured active channel width to widths extracted from lidar topography show a high degree of correlation, indicating that lidar-derived width measurements are a viable alternative when field measurements are difficult to obtain (Figure 4).

The effect of slope on sediment storage also reflects the interaction between the LWD and channel morphology. There is no relation between slope and all measurements of deposit volume. However, a more narrow analysis of the stored deposit volume $> 10$ m$^3$ shows deposit volume increases as channel slope decreases. This indicates that small amounts of sediment can be retained and stored regardless of the channel slope. However, larger deposits accumulated on shallow slopes. Consequently, energy dissipation on shallow slopes may help to encourage more deposition behind LWD than on steep slopes where flow energy is higher, and slope may be a useful predictor of LWD sediment storage.


       The volume of deposition at the terminal fan for each basin outlet modeled using Equation 2 overpredicted the fan deposit by several orders of magnitude. The Gartner et al. (2014) volume model has been shown to be successful in the region where it was developed (the Transverse Range of southern California) and where a large contribution of sediment is derived from hillslope erosion (Rengers et al., 2021), but it overpredicts at this study site. In contrast, Equation 3 developed by Pelletier and Orem

(2014) predicted sediment volumes at the terminal fan that were closer to the observed volumes. This might be because that model was developed in a similar region of New Mexico, with a similar elevation/climate (both at 2500-3000 m) and lithology (rhyolite). The forced storage of debris flow sediment by LWD in the Tadpole study area retained a larger or comparable volume of sediment as was observed at the terminal fan of the basin outlets, which may explain some deviations from the Pelletier and Orem (2014) volume model. Moreover, the Tadpole study area has steep slopes on Tadpole Ridge that rapidly

decrease at drainage areas of less than 1 km$^2$ (Table 1), but the region studied by Pelletier and Orem (2014) maintained steady slopes and channel scour at larger drainage areas prior to deposition (greater than 1 km$^2$). Therefore, the total volume observed in their model may be calibrated on observations of more sediment scour. Consequently, regionally calibrated empirical models may be the best approach for regional volume predictions, but local influences of site geomorphology may add to variability in predictions versus observations.

Our greenstick analysis of wood breakage helps to quantify the flow threshold where LWD may no longer have a substantial effect on retaining sediment. As flows become larger and increase in velocity, LWD will break or move and retain fewer debris flow deposits. The presence of unbroken wood pinned against trees after the debris flow events implies that those wood pieces did not experience stresses in excess of yield strength. Our wood-breaking velocity analysis using Equation 9 agrees well with field observations. Some of the wood breaking velocities for selected LWD in relevant geometries are lower than the observed peak velocity at the geophones of 4 m/s (Figure 10). These deviations likely reflect potential for velocity differences across

channels Tad-1 to Tad-4, as well as changes in channel morphology that could reduce flow speed such as wide channels or lower slopes. Despite these uncertainities, overall the predicted velocities generally correspond with our expectations of wood



larger than a breaking velocity of 4 m/s in the narrow confined channels, and less than the breaking velocity at wide channel sections or near the basin outlet where the channel becomes unconfined. Therefore, this approach may be a potential tool for
estimating debris flow velocities, which could be used to constrain model simulations.

In summary, this study indicates that in regions where there is a potential for substantial LWD interaction with debris flow sediment, the LWD may strongly control the overall location of sediment deposition and alter predictions of deposit volume. The amount of sediment stored behind LWD exceeded the sediment deposited in the terminal fan in three out of four cases. In situations where the debris flow momentum was smaller than the breakage capacity of the wood, we saw that LWD can
substantially influence debris flow sediment storage. However, in larger debris flows, LWD may not be sufficient to retain sediment within the channel before reaching a terminal deposition zone (Booth et al., 2020). Consequently, the effect of LWD on sediment storage will be dependent on the rainfall rate (Gartner et al., 2014), which ultimately controls the debris flow size and watershed characteristics, such as channel width variations. In addition, debris flow sediment stored by LWD may periodically load channels with sediment, potentially leading to more extreme responses and downstream sedimentation during
future storms when this sediment is mobilized. At this study site, sediment stored behind wood with diameters exceeding 10 cm may remain in channels and be available for future debris flows because of the slow decay rate for wood with D> 10 cm (Harmon and Sexton, 1996). These observations give a snapshot of the influence of LWD for an observed set of rainfall and watershed characteristics. More work would be beneficial to develop a framework to model the potential storage as a function of rainfall intensity, stem density, drainage area, and channel width.

**6  Conclusions**

Large woody debris (LWD) is often entrained and transported during debris flows. In some cases the LWD can interact with the flow to retain sediment in channels, which influences predictions of debris flow volume expected at channel outlets. In this study we observed that debris flow sediment retention in steep headwater streams was dependent on both the characteristics of the LWD and the channel morphology. LWD with larger diameters retain more sediment, and the length of LWD with
respect to channel width strongly controls sediment retention. The largest deposits were found at the lowest channel slopes; however, LWD retained small volumes of debris flow sediment regardless of the overall channel slope. Future predictions of the location of debris flow sedimentation in small headwater streams could be achieved by estimating a characteristic wood length, and identifying areas where the ratio of wood length to channel width are between 0.25 and 1. Additionally, we found that observations of LWD dimensions sufficient to hold back sediment without breaking may be a useful future tool for estimating
debris flow velocity, and this may be useful for eventually helping to determine thresholds below which LWD may influence deposit volumes.





**Figure 1.** (a) Study area within the Tadpole wildfire perimeter. LWD and/or debris flow deposit measurement locations within four watersheds (Tad-1 to Tad-4) are identified. The size of the dots on the map scales between the minimum ($0 \, \mathrm{m}^3$) and maximum volume ($208 \, \mathrm{m}^3$) observed at each LWD/debris flow measurement location. (b) Inset map showing the location of the study site within the state of New Mexico, USA. (c) Gartner et al. (2014) Model predictions of debris flow volume using the maximum rainfall intensity observed at the geophone location. Note that the extent of (c) is approximately the same as (a).





**Figure 2.** Schematic showing the geophone setup. (a) Photo of the rainfall-triggered geophone setup. Note that the downstream geophone location is out of view. (b) Plan view dimensions of the channel, channel banks, and geophones. (c) Cross-sectional view of channel and geophone location. Photo Credit: F. Rengers.




**Figure 3.** Photographs showing each LWD class. Dashed lines help to identify the wood pieces, and arrows indicate the direction of flow. Photo Credit: F. Rengers.



**Figure 4.** Analysis of channel width extracted from the pre-event digital elevation model and field measurements. (a) Blue line shows one of the cross-sections extracted from the lidar DEM. The cross-section has been centered so that the thalweg is located at 0 m, and the distance away from the thalweg is shown as either positive or negative values. Dashed lines were automatically determined at the location on each bank that is 1 m above the thalweg. Arrows indicate the channel width measurement location. (b) Plot of field measurements of channel width versus measurements obtained automatically from the lidar data. (c) Channel width with respect to drainage area, note semi-log scale.



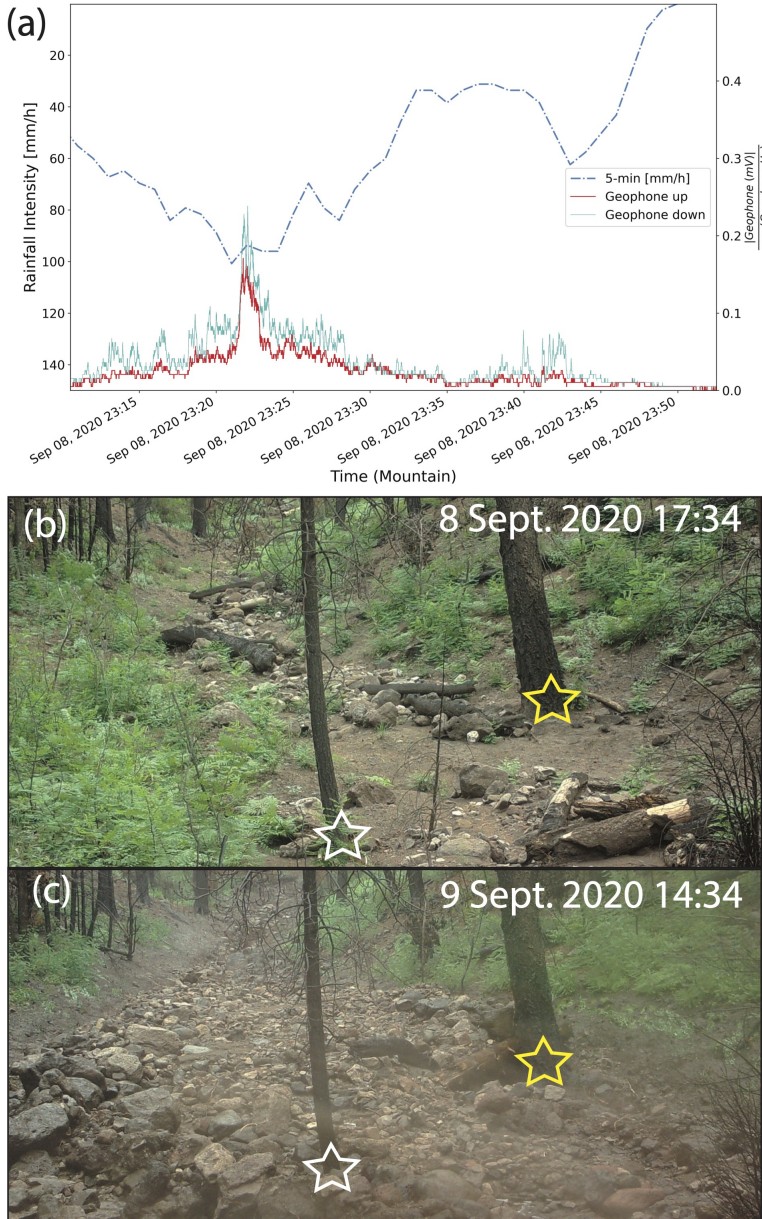

**Figure 5.** (a) Rainfall intensity and corresponding geophone response during a debris flow. "Geophone up" represents the upstream geophone, and "Geophone down" represents the downstream geophone. Note that the rainfall starts at the beginning of the geophone time period. In order to calculate the intensity values over the period of minutes ($\mu$), the total rainfall must be summed over $\mu$ and thus an intensity is not plotted for that duration. This explains the lag in the rainfall intensity lines compared to the geophone lines. (b) Photo of the channel reach near the geophones on 8 Sept 2020 at 5:34pm (local time). Photo Credit. L. McGuire. (c). Photo of the channel reach near the geophones following the debris flow at 9 Sept 2020 at 2:34pm (local time). Stars indicate the same location in each photo. Photo Credit. L. McGuire.





**Figure 6.** (a) The total volume of sediment normalized by the total number of deposits within a class. (b) The total volume of sediment stored behind each measured wood category. (c) Histogram showing the number of LWD pieces in each class. (d) The total volume of sediment stored behind either buried or jam LWD classes normalized by the total number of deposits within 10 cm diameter bins. (e) Total volume of sediment trapped in deposits with either buried LWD or wood within jams. The maximum diameter of the LWD is binned in 10-cm intervals. (f) Histogram showing wood-diameter distribution in the dataset.



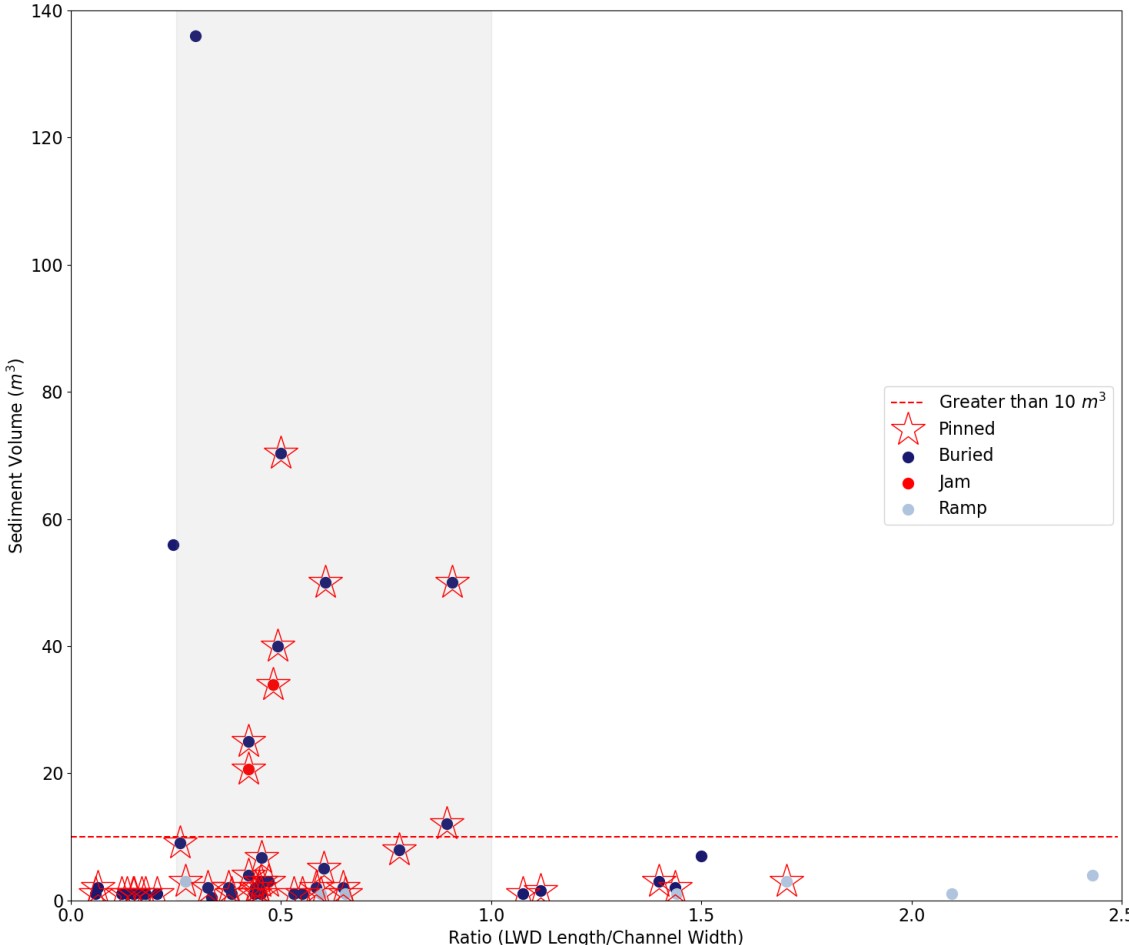

**Figure 7.** A comparison of the ratio of the LWD length to channel width $\zeta_{LW}$ at 1 m above the channel bottom versus the trapped deposit volume. The color of each point is based on the class of the LWD. The bridge LWD class was removed because that wood did not retain any sediment. The loose LWD class was removed because that wood did not actively restrict sediment movement downstream. The shaded gray region denotes the ratio $\zeta_{LW}$ values associated with the maximum sediment retention volumes. Dashed line separates deposits greater than and less than 10 m$^3$.





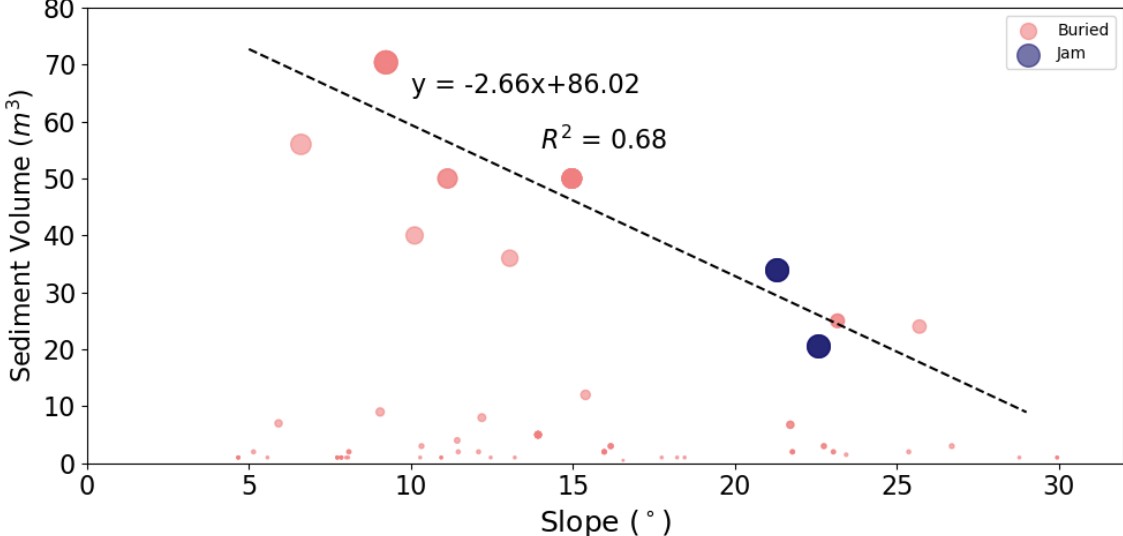

**Figure 8.** Slope angle (degrees) versus the deposit volume. The points are scaled by deposit volume size, and a regression is fit to all deposits larger than $10\,\mathrm{m}^3$.



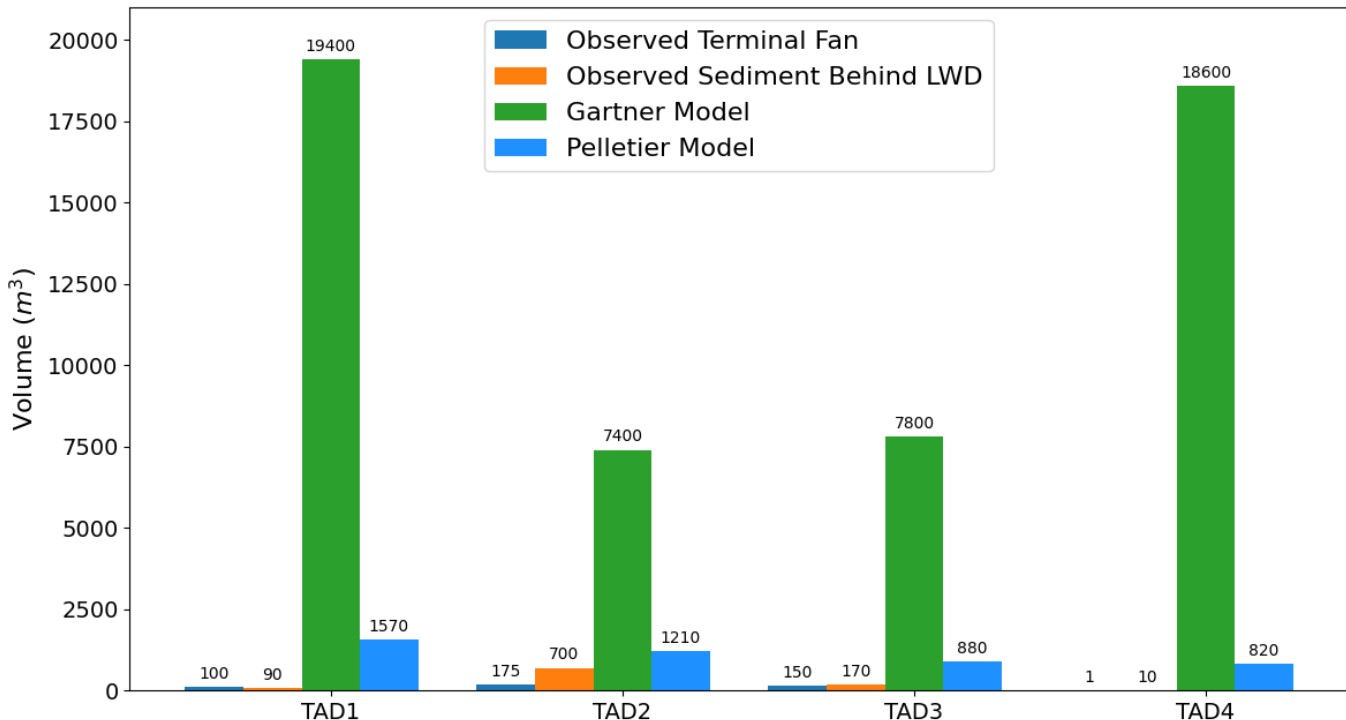

**Figure 9.** Comparison of sediment stored behind LWD with sediment stored in terminal fans at the basin outlet. In addition, model estimates (Gartner and Pelletier) of post-wildfire sediment volumes at the basin outlet are displayed. The volume of observed and modeled volumes are shown above each bar with units of $\mathrm{m}^3$.



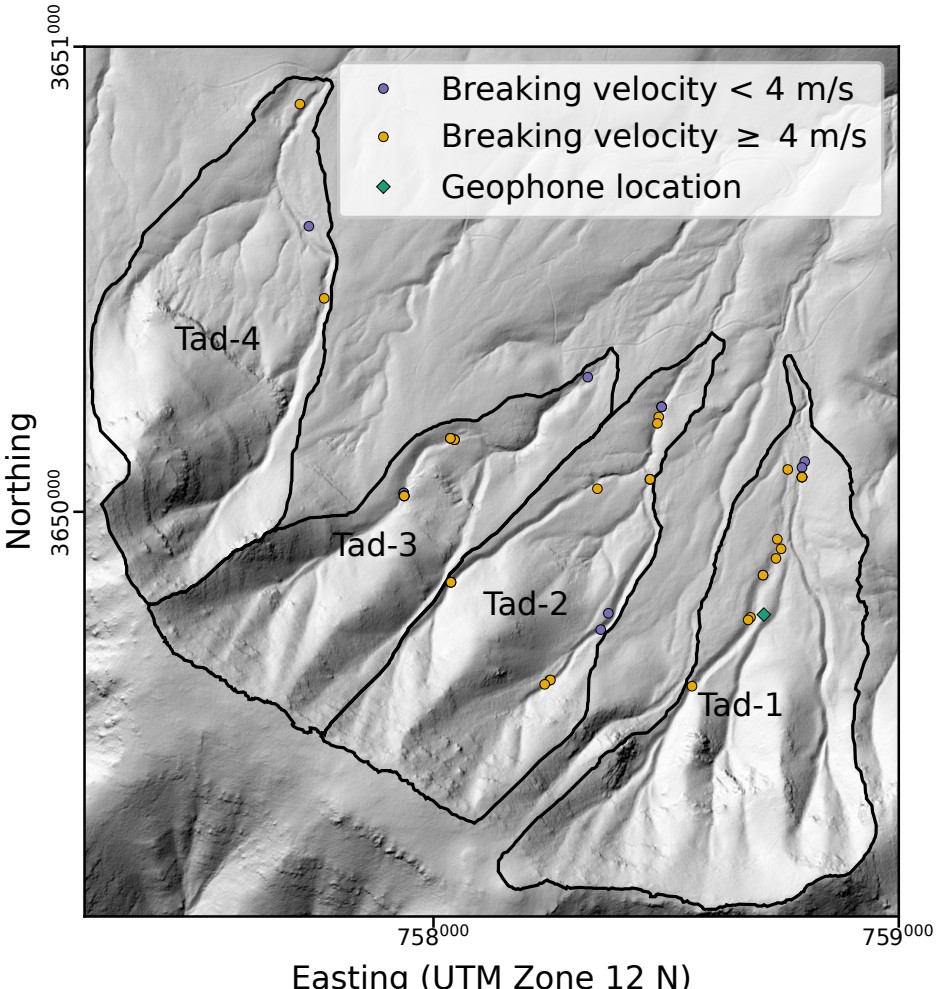

**Figure 10.** Map of estimated velocity based on the wood dimensions. The velocity (4 m/s) is used as a threshold because it was the maximum velocity measured by the geophones.





**Table 1.** Watershed Characteristics. The deposit volumes for each watershed are shown as either trapped by LWD, or unconstrained (i.e., the deposit stops without any interference by LWD). Note that burn severity for each watershed is labeled as: Low, Moderate (Mod), and High. Watershed locations are displayed in Figure (1).

|  | Tad-1 | Tad-2 | Tad-3 | Tad-4 |
|---|---|---|---|---|
| Area (km$^2$) | 0.43 | 0.32 | 0.29 | 0.39 |
| Average Slope (°) | 24.1 | 23.7 | 24.3 | 22 |
| Relief (m) | 340 | 330 | 350 | 350 |
| % Low | 0.16 | 0.24 | 0.14 | 0.49 |
| % Mod-High | 0.84 | 0.76 | 0.86 | 0.51 |
| Date of Largest Debris Flow Deposit | 8 Sept. 2020 | 18 Jul. 2020 | 21 Jul. 2020 | 21 Jul. 2020 |
| Max Fan Vol. at Outlet (m$^3$) | 100 | 150-200 | 150 | No Fan |
| Total LWD Trapped Volume (m$^3$) | 87 | 700 | 170 | 12 |
| Total Unconstrained Volume (m$^3$) | 100 | 505-555 | 150 | 1 |
| Ratio LWD Trapped Volume to Fan Volume | 0.9 | 4 | 1.1 | 12 |



**Table 2.** Storm Responses Following the Fire

| Date | RG I15 (mm h$^{-1}$) | RG and Geophones I15 (mm h$^{-1}$) |
|---|---|---|
| 18 July 2020 | 53 | 16.8 |
| 21 July 2020 | 52 | 24.0 |
| 22 July 2020 | n/a | 12.8 |
| 24 July 2020 | 27 | 34.4 |
| 25 July 2020 | 25 | 35.2 |
| 26 July 2020 | 21 | 12.8 |
| 28 July 2020 | 31 | 39.2 |
| 23 August 2020 | 7 | 19.2 |
| 24 August 2020 | no rainfall | 16.8 |
| 1 September 2020 | 18 | 19.2 |
| 8 September 2020 | 93 | 86.4 |



*Data availability.* Data used for the analyses in this study are available in (Rengers et al., 2022a, b).

*Author contributions.* Rengers, Kean, McGuire, and Youberg developed field instrumentation design. Cadol designed field survey strategy. McGuire, Rengers, Youberg, Gorr, Hoch, Barnhart, and Beers helped with field data collection and collation. Barnhart led wood-break velocity analysis. Rengers prepared the manuscript and performed data analysis with contributions from all co-authors.


*Competing interests.* The authors declare no competing interests

*Disclaimer.* Any use of trade, firm, or product names is for descriptive purposes only and does not imply endorsement by the U.S. Government.

*Acknowledgements.* This work was supported in part by the U.S. Geological Survey Landslide Hazards Program.



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
