# Peer review of "The Influence of Large Woody Debris on Post-Wildfire Debris Flow Sediment Storage"

_EGUsphere, 2022_

## Author Response (AR1)

**Responses to reviewers are shown below in bold.**

Reviewer 1:

Revision of the manuscript number "egusphere-2022-1398" entitled "The Influence of Large Woody Debris on Post-Wildfire Debris Flow Sediment Storage". The paper presents valuable field-measured information of woody-laden debris flows. It can be accepted for publishing in Natural Hazards and Earth System Sciences at the present form. It provides an adecuate analysis of the information. I only have a few comments that must be addressed before its publication.

**1.** P3_L65. "...the dominant soils on the site are Mollisols, Enceptisols, and Alfisols (U.S. Forest Service, 2020)." Soils should be defined based on their textural descriptions, as this gives a better sense of their composition mechanically. Even better if you have measured some physical parameters, e.g., friction angle, cohesion, or any other mechanical variable in this region, provide such information. It will be more valuable for future works based on mathematical modelling and numerical simulation.

**We have added this additional text, but unfortunately we don't have other parameters such as friction angle or cohesion:**

**"...and the grain size suggests a loam texture (43% sand, 45% silt, 12% clay)"**

**2.** P6_L165. "...stands between 2740 and 3040 m, and Engelmann spruce (Picea engelmannii) and corkbark fir (Abies lasiocarpa var. arizonica) > 3040 m." Correct "m" to "m.a.s.l." in these cases.

**Changed**

**3.** P7_Eq5. Please cite the previous works where this equation has been used for similar or same purposes, e.g.,:

*Abbe, T. B., & Montgomery, D. R. (1996). Large woody debris jams, channel hydraulics and habitat formation in large rivers. Regulated Rivers: research & management, 12(2-3), 201-221.

*Manners, R. B., Doyle, M., & Small, M. J. (2007). Structure and hydraulics of natural woody debris jams. Water Resources Research, 43(6).

**Great suggestion. We have now added those references, and were previously unaware of the use of the equation in that literature. Thanks!**

**4.** P7_Eq5_below. Clarify this if appropriated: "We used a weighted average density of $\rho=1680$ kg/m³"

**Change to say "We used a weighted average density of $\rho=1680$ kg/m$^3$"**

**5.** P7_Eq9. Appropriately describe where is this equation derived from or cite the paper where this was taken from, if so.

**Good point. This equation is the consequence of combining multiple prior equations. The text was revised to clarify how it was generated. Here is the revised text:**

**"We calculated the value of u at yield by combining Equations 4, 6, and 7, and rearranging for u."**

**6.** P12_Fig1. Improve these maps, their scales and labels. The current "map b" is not actually improving the location insight of the studying place, please improve this map.

**This is a useful suggestion. I mostly focused on panel b. For panel a, we have a scale, label, and tick marks that indicate the location. For b, I added a shaded relief map to show the location of the mountain ranges. I tried adding in a map of the entire USA to show the location of New Mexico in that context, but it didn't fit very well. The new shaded relief shows the locations of topography, and I also added in labels to indicate the capital city, Albuquerque, and a hollow star to indicate the study site. A scale bar is also added to show the extent. With this new information in panel b, I think readers will be able to understand the location in New Mexico, USA that we are working, and they can independently use the lat/long in panel A to find the spot using something like google earth. Panel c has a legend, and I added a scale bar. In the caption we also note that the location is the same as in panel a.**

Reviewer 2:

Overview:

This manuscript provides an examination of the role of LWD on sediment storage in post-wildfire catchments. A field investigation of watershed morphology with surveys of sediment retention and LWD characteristics was undertaken. These data were used to examine the interactions between LWD and sediment retention. A method to back-analyze debris-flow velocity from estimated wood breakage is also presented.

General comments:

This manuscript is of interest to researchers and practitioners working with post-wildfire debris-flow hazards. The data on sediment yield and sediment storage for the watersheds studied is valuable, and as the authors show, highlights the importance of collecting region-specific data. The work presents a novel application of LWD field surveys to infer flow dynamics.

Specific comments:

Line 19:  Is "ecological services" a standard term? "Processes" or "functions" seems more appropriate to me.

**This is a term typically used to describe the economic benefits of properly functioning ecological systems:**

**https://en.wikipedia.org/wiki/Ecological_goods_and_services#:~:text=Ecological%20goods%20and%20services%20(EG%26S,rather%20than%20to%20humans%20alone.**

**But since this paper is not focused on economics, we changed it to 'ecological processes' for clarity.**

Lines 25 – 26: I recommend changing the order of the sentence to go from causes to effects, "Processes such as root weakening, wind throw and disease are enhanced in forests burned by wildfire, which accelerates the introduction of wood into channels."

**Changed.**

Line 75: Please state the significance of the San Gabriel Mountains for readers who are not familiar with previous study locations within the US where post-wildfire debris flow work has been completed.

**Good suggestion. We have now added several references to show the prior debris flow observations in the San Gabriel Mountains.**

Line 76: Please provide a brief description of chapparal vegetation.

**Changed to add this sentence: "The San Gabriel Mountains are primarily vegetated by chaparral plants, which are sclerophyllous woody shrubs found in semi-arid environments that are prone to burn every 30-150 years (Halsey, 2005).**

Lines 91 – 92: Provide a reference for the geophone cross-correlation method used to estimate the flow velocity.

**I added a reference to Kean et al. 2015, who use a similar approach.**

Line 115: Is the reported lidar point density bare-earth points?

**Yes. Changed to say 'ground point density' since they are technically ground points.**

Line 129: This sentence should be rewritten to either state how volume was related to the ratio of wood length to channel width, as stored sediment volume does not appear in (1).

**Changed to say: "Prior work has recognized that LWD deposition in a channel is related to the length of the LWD (L) versus the channel width (W) (Vaz et al., 2013). Herein we define this ratio as:**

**ζLW = L/W (1)**

**We examined the volume of debris flow sediment stored behind LWD with respect to ζLW ."**

Line 157: What are the units for the average burn severity, B?

**Good catch. I've now changed the sentence, so it reads:**

**where Yp is sediment yield in mm, S is average basin slope (m/m), B is average soil burn severity, a = 1.53, b = 1.6, and c = 1.7. The categorical burn severity variables are converted to the following unitless values: low = 1, moderate = 2, high = 3.**

Line 174: Is the assumption of greenstick fracture behavior based on the incomplete combustion of the stems and the relative short duration following the fires? Are there scenarios where a different fracture behavior would be assumed? This may be a point for the discussion section.

**We think that this would always be the type of failure of wood regardless of the burned state or amount of wetness. This is simply the name of the type of fracturing that the material undergoes, it is the same regardless of amount of burning or wetness/dryness. This type of fracturing occurs in many materials from bones to wood.**

**We changed the sentence to say:**

**"Therefore, we related wood size to breaking velocity, which is the velocity of flow required to break wood, assuming greenstick fracture behavior as the failure mechanism for both unburned and partially burned wood (Ennos and Van Casteren, 2010).**

Line 175: Debris flow should be hyphenated preceding event.

**I mostly agree with this, but I've seen different journals handle it differently and it comes down to the journal house-style. The JGR style is not to hyphenate at all. USGS style suggests that you hyphenate when the term "debris flow" is modifying, as in this case "a debris-flow event". I have removed hyphens from the entire manuscript, and am waiting for guidance on the EGU style.**

Lines 244 – 245: Were there any observations of broken wood within the study area?

**Almost all the wood was broken in some manner because it was never an intact tree. But I think you are asking if we saw a 'smoking gun' that suggested that wood broke in the channels. I'd say we mostly didn't see that, but that also agrees with our calculations, which suggest that the velocities were rarely greater than the breaking capacity and that is why we saw wood jamming up and storing sediment.**

Line 263: As it's stated, this is suggesting that there is information on how the LWD affects the velocity, however, it would be more accurate to say the LWD can be used to provide estimates of velocity.

**Changed sentence to say: "Field data combined with modeling are used to understand how LWD influences debris flow volume storage, and how LWD can be used to estimate flow velocity."**

Lines 267 – 278: This sentence needs to be edited to more clearly state that given the length of the fire, trees with diameters less than 10 cm would likely be fully consumed.

**Changed to say: "The majority of wood diameters measured were greater than 10 cm, possibly because wood of smaller diameters was destroyed by the fire. The total length of fire at any location is unknown, however, 10-100 hour fuels are (2.5-7.6 cm) and it is likely that wood with diameters less than 10 cm were fully consumed."**

Lines 282 – 284: The discussion on field measurements seems out of place in this paragraph, perhaps move this information to the paragraph in lines 265 – 271.

**I'm not sure that would make sense. The topic sentence of this paragraph indicates that it is about channel morphology, and the prior sentences focus on channel width (a type of channel morphology). This sentence is just closing the loop on lidar-extracted channel width versus measured channel width to indicate that it is ok to use lidar extracted channel width. The paragraph starting on line 265 is entirely about the role of wood characteristics on sediment volume. They seem to be very different topics.**

Line 315: Do the authors suspect the idealization of the tree stems as cylinders could influence the results in general? As trees thin towards the top, could preferential breaking where the stem is thinner affect these analyses?

**The analysis of breaking velocity assumes that tree will break at the location with the greatest bending moment. For a cylinder pinned at both ends or in the middle and subjected to a uniform force, this occurs at the midpoint. I am not sure how different the results would be if a truncated cone were used instead of a cylinder. I suspect that the results would be more sensitive to relaxing the assumption of a uniform force along the length of the tree (e.g., considering impacts by individual entrained sediment clasts) than to relaxing the assumption that the tree is a cylinder. I think this because the observed pinned trees had relatively little variation in diameter along their lengths. No change made to the text.**

Figure 4: x-axis labels are missing on panels (a) and (b).

**Thanks for catching that, it is fixed now.**

Figure 5(a): Should the geophone units on the right y-axis be mV/mV, thus dimensionless?

**This is a good catch. I changed the right y-axis so that the units (mV) are shown in both the numerator and denominator in fraction on the label.**

Figure 6(f): y-axis labels are not the same size as the other panels.

**Changed so 6f now has the same label size. Thanks. Also, I want to note that I changed the caption a bit, and reran my analysis to be a little bit more specific. I noticed that (a) was actually plotting the sediment volume/number of wood pieces instead of what was labeled "Sediment Volume per Deposit". So I fixed it to be the measured sediment volume divided by the number of deposits. I also changed the caption for c and the label, because I want c to reflect the number of LWD pieces. Finally, I tried to make the entire caption for this figure clearer.**

Figure 7: Remove "greater than" from the label for the red dashed line. The ς symbol is used in the caption, but ratio of LWD length to channel width is used on the axis label. I would recommend being consistent with one or the other.

**Removed 'Greater than' and changed the x label to use the symbol.**

**Finally, I wanted to note a change in figure 9. It came to our attention that the volume in Tad2 accidently included three spots where sediment was stored in the watershed, but not held back by LWD. These spots are shown as Non-LWD deposits in Figure 1. Removing that data, lowered our overall estimate of sediment stored. Similarly, a few deposits were missing from Tad4, raising that volume stored behind LWD. We apologize for this errata.**